# The Influence of Probiotic Supplementation on Depressive Symptoms, Inflammation, and Oxidative Stress Parameters and Fecal Microbiota in Patients with Depression Depending on Metabolic Syndrome Comorbidity—PRO-DEMET Randomized Study Protocol

**DOI:** 10.3390/jcm10071342

**Published:** 2021-03-24

**Authors:** Oliwia Gawlik-Kotelnicka, Anna Skowrońska, Aleksandra Margulska, Karolina H. Czarnecka-Chrebelska, Igor Łoniewski, Karolina Skonieczna-Żydecka, Dominik Strzelecki

**Affiliations:** 1Department of Affective and Psychotic Disorders, Medical University of Lodz, 92-216 Lodz, Poland; anna.zabka@stud.umed.lodz.pl (A.S.); dominik.strzelecki@umed.lodz.pl (D.S.); 2Admission Department, Central Teaching Hospital of Medical University of Lodz, 92-216 Lodz, Poland; aleksandra.margulska@gmail.com; 3Department of Biomedicine and Genetics, Medical University of Lodz, 92-213 Lodz, Poland; karolina.czarnecka@umed.lodz.pl; 4Department of Biochemical Sciences, Pomeranian Medical University in Szczecin, 71-460 Szczecin, Poland; sanprobi@sanprobi.pl (I.Ł.); karzyd@pum.edu.pl (K.S.-Ż.)

**Keywords:** depression, metabolic syndrome, probiotics, microbiota, inflammation, oxidative stress

## Abstract

There is a huge need to search for new treatment options and potential biomarkers of therapeutic response to antidepressant treatment. Depression and metabolic syndrome often coexist, while a pathophysiological overlap, including microbiota changes, may play a role. The paper presents a study protocol that aims to assess the effect of probiotic supplementation on symptoms of depression, anxiety and stress, metabolic parameters, inflammatory and oxidative stress markers, as well as fecal microbiota in adult patients with depressive disorders depending on the co-occurrence of metabolic syndrome. The trial will be a four-arm, parallel-group, prospective, randomized, double-blind, controlled design that will include 200 participants and will last 20 weeks (ClinicalTrials.gov identifier: NCT04756544). The probiotic preparation will contain *Lactobacillus helveticus* Rosell^®^-52, *Bifidobacterium longum* Rosell^®^-175. We will assess the level of depression, anxiety and stress, quality of life, blood pressure, body mass index and waist circumference, white blood cells count, serum levels of C-reactive protein, high-density lipoprotein (HDL) cholesterol, triglycerides, fasting glucose, fecal microbiota composition and the level of some fecal microbiota metabolites, as well as serum inflammatory markers and oxidative stress parameters. The proposed trial may establish a safe and easy-to-use adjunctive treatment option in a subpopulation of depressive patients only partially responsive to pharmacologic therapy.

## 1. Introduction

Metabolic syndrome (MetS) and depression are among the most common and debilitating disorders worldwide [1]. Moreover, depressive disorders (DDs) often coexist with MetS, thus further increasing mortality risk [2,3]. A meta-analysis [4] showed that individuals with depression had 1.5 times higher odds of developing MetS, which is diagnosed in 30% of depressed subjects [5]. Although the exact mechanisms underlying this association are poorly known, the high co-occurrence rate of DDs and MetS suggests a possible pathophysiological overlap. Among others, altered microbiota has been suggested as one of the factors [1,3,5,6,7,8,9]. The term microbiota refers to trillions of organisms present in the human body. They have a substantial impact on health, including the human immune status [10,11]. The diversity and stability of the microbiota are important indicators of the overall health condition of an individual [12]. The intestinal microbiota has been shown to be an essential part of the bidirectional and complex gut–brain axis [13,14]. However, the most commonly studied fecal microbiome is different from the whole microbiome of the gastro-intestinal (GI) tract [15] The diversity of the gut microbiota has emerged to play a significant role in the occurrence of mood and anxiety disorders [12,16,17,18,19,20,21,22]. In a recent systematic review and meta-analysis of observational studies [20], it was shown that several taxa—specifically, family *Prevotellaceae*, genus *Corprococcus*, and *Faecalibacterium*—were decreased in major depressive disorder (MDD) patients compared to non-depressed controls. Moreover, results of another study indicate that depression and anxiety severity were negatively associated with bacterial richness and α-diversity [22]. Moreover, there are a lot of scientific data on the implication of the intestinal microbiota function in the etiopathogenesis of MetS and the connection of dysbiosis with abdominal obesity [23,24,25]. Overall, most studies until today have demonstrated a reduction in gut microbiome diversity in obese subjects compared to lean ones; additionally, it has been shown that the ratio of Firmicutes/Bacteroidetes (F/B ratio) was higher in obese and overweight individuals, particularly with metabolic complications [25]. These findings are consistent with animal models of alterations of gut microbiota, e.g., the absence of gut microbiota in germ-free (GF) mice resulted in decreased immobility time in the forced swimming test relative to healthy control mice. Furthermore, fecal microbiota transplantation (FMT) of GF mice with microbiota derived from MDD patients resulted in depression-like behaviors [17]. The complex net involving depression, metabolic diseases, inflammation, oxidative stress (OxS) parameters and microbiota has been recently discussed in another paper [26].

Certain microbiota interventions, e.g., introduction of the Mediterranean-style diet, intake of fermented food and dietary fiber, a plant-based diet, and consumption of prebiotics (oligosaccharides, fructans) may reduce the risk of developing depression [27], as well as MetS and its after-effects, e.g., chronic kidney disease (CKD) or cardiovascular diseases (CVDs) [28,29]. Furthermore, recent trial reports on using probiotics (live microorganisms that, if consumed in adequate amounts, bring the host health benefits [30]) demonstrated their usefulness in depression or anxiety outcome measures [31,32,33,34,35,36]. The results of the most recent meta-analysis on the use of probiotics in clinical depression demonstrated that probiotics are effective in reducing depressive symptoms when administered as an add-on; however, they do not seem to offer significant benefits when used as stand-alone treatment [36]. It has been suggested that the microorganisms can form a new group of drugs named “psychobiotics” [37], which makes future research in the area very promising. Moreover, the main findings of the recent trials and reviews are that some of the selected probiotics can discretely improve some of the clinical components of the MetS, as well as some inflammatory biomarkers associated with the syndrome [38,39,40]. Despite the fact that, according to a recent systematic review [40], probiotics intake in patients with MetS resulted in improvements in body mass index (BMI), blood pressure (BP), glucose metabolism, and lipid profile in some studies, these beneficial effects seem to be clinically non-relevant compared to standardized therapy. To sum up, holistic lifestyle changes may be necessary as an adjunct to probiotics treatment to provide long-term improvements. It has also been highlighted in the literature the advantage of function-based rather than taxonomy-based strategies for the selection of candidate strains for the next-generation probiotics [41].

The primary aim of the study is to assess the effect of probiotic supplementation on diminishing symptoms of depression in adult patients diagnosed with depressive disorders depending on the co-occurrence of MetS. The secondary aims are to assess the effect of probiotic supplementation on anxiety and stress symptoms, metabolic parameters, inflammation and oxidative stress markers, short-chain fatty acids (SCFAs) level, and microbiota composition in feces in these patients. The preliminary aim is to compare the markers of fecal microbiota function and composition, as well as inflammation and OxS parameters between depressed patients with and without MetS.

The study hypothesis is that comorbidity of depression and MetS increases the probability of therapeutic effect of probiotic supplementation in the case of depressive and anxiety symptoms via the increase of α-diversity in fecal microbiota and thus the synthesis of SCFAs. If it is true, this may help to establish a subpopulation of patients vulnerable to positive response to probiotic supplementation. Additionally, along with the evaluation of inflammatory and oxidative status parameters, this could determine potential biomarkers of therapeutic response among depressive patients.

## 2. Materials and Methods

### 2.1. Design

The trial will be a four-arm, parallel-group, prospective, randomized, double-blind, controlled design that will last 20 weeks (eight weeks for an intervention period) (Figure 1). The trial was registered in the clinicaltrials.gov registry (ClinicalTrials.gov identifier: NCT04756544). PRO-DEMET study protocol design and publication have been planned according to CONSORT 2010 checklist. 

### 2.2. Patients

The study population will consist of 200 patients recruited in the Central Teaching Hospital (psychiatric inpatient and outpatient clinics) of the Medical University of Lodz, Poland, other psychiatric outpatients clinics in Lodz, and through advertisements in social media. All the subjects will be diagnosed with depressive disorders according to the upcoming International Classification of Diseases (ICD-11) [42] and half of them with accompanying MetS will be diagnosed according to International Diabetes Federation (IDF) criteria [43]. 

As for the sample size, our approach is as follows. An online tool (https://www.gigacalculator.com/calculators/power-sample-size-calculator.php, accessed on 2 December 2020) estimated that an overall group of 195 individuals (39 per group) is required to detect possible intergroup differences. After a preliminary analysis of variance (ANOVA), we will conduct post hoc tests to ascertain the exact differences.

Ideally, for post hoc analyses the population should be larger than 65 people per group for Montgomery-Åsberg Depression Rating Scale (MADRS) score comparisons, 74 per group for short-chain fatty acids (SCFAs) level comparisons and 84 per group for α-diversity comparisons; however, with 40 persons per group it still seems like it will be possible to detect a statistical difference between groups regarding all the above outcomes. The details of sample size calculations are described in Appendix A section.


**Randomization**


The patients will undergo randomization stratified according to MetS presence and will be allocated to probiotic or placebo treatment in a 1:1 ratio by an independent researcher. The trial participants, care providers, outcome assessors, and data analysts will be blind to study group assignment. Unblinding will be permissible only if any serious adverse events occur during the course of the trial. Randomization will be computed using a computer-based random number generator (https://www.randomizer.org/, accessed on 8 March 2021).

The study groups will be as follows:PRO-DMS: probiotic + depression + MetSPLC-DMS: placebo + depression + MetSPRO-D: probiotic + depressionPLC-D: placebo + depression

Each patient will receive information on the purpose and structure of the investigation. Deliberate written and informed consent to take part in the study will be required. Anyone who fails to meet the inclusion criteria will not qualify for participation in the project. The participants will be able to opt out of the study at any time before or during implementation of the protocol. The data collected and samples will be pseudonymized for the purposes of future use, and the informed consent form will include information on the future use of samples, material, and personal data. The study will be conducted in accordance with the Declaration of Helsinki, and the principal study investigator (PSI) applied to the Bioethics Committee of the Medical University of Lodz for an approval of the whole protocol. 

The study timeline will consist of the following appointments (Figure 2):V0, “recruitment visit”, preferably an online formula: assessment of the inclusion and exclusion criteria, the study questionnaire (SQ) and the MADRS completion, informed consent, full psychiatric examinationV1 (no longer that five days after V0), “randomization visit”: Depression, Anxiety, Stress Scale (DASS), The World Health Organization quality of life-BREF (WHOQOL-BREF) completion, blood pressure (BP), body mass index (BMI), waist circumference (WC) measurements, blood and stool collectiont_1_–t_3_: personal, telephone, or e-mail monitoring every two weeks according to the monitoring questionnaire (MQ)V2, “the end of the study visit”, eight weeks after V1: MQ, MADRS, DASS, WHOQOL-BREF completion, BP, BMI, WC measurements, blood and stool collectionV3, “a follow-up visit”, 12 weeks after V2: SQ, MADRS, DASS, WHOQOL-BREF completion, BP, BMI, WC measurements

To be qualified for the trial, the subjects will have to meet all the inclusion criteria and none of the exclusion criteria, as stated in Table 1.

### 2.3. Intervention

At the beginning of the study (V0), the participants will be requested to follow their routine physical activity, dietary intakes, or any other lifestyle activities throughout the whole trial. These evaluations will be performed by means of the study-specific self-assessment Monitoring Questionnaire (MQ). The probiotic groups (PRO-DMS and PRO-D) will receive one capsule containing the probiotic mixture powder in the amount of 3 × 10^9^ colony forming units (CFU) until end of shelf life. The probiotic preparation will contain two bacteria strains, i.e., *Lactobacillus helveticus* Rosell^®^-52, *Bifidobacterium longum* Rosell^®^-175, and excipients, i.e., potato starch, magnesium stearate, and the capsule shell of hydroxypropyl methylcellulose (Sanprobi Stress^®^, Sanprobi Sp. z o. o., Sp. k., Szczecin, Poland; probiotic powder manufacturer—Institute Rosell-Lallemand, Montreal, Canada). The placebo groups (PLC-DMS and PLC-D) will receive the same capsule containing only the excipients, i.e., maize starch, maltodextrins, and the capsule shell (manufacturer—Sanprobi Sp. z o. o., Sp. k., Szczecin, Poland). The placebo will be indistinguishable in color, smell, and taste from the probiotic formulation. The participants will be given the supplements before breakfast. 

### 2.4. Outcome Measures

The primary outcome measure will be the Montgomery-Åsberg Depression Rating Scale (MADRS). The secondary outcomes measures will include the Depression, Anxiety, Stress Scale (DASS) with subscales scores, quality of life level assessed with the WHOQOL-BREF instrument, blood pressure (BP), body mass index (BMI) and waist circumference (WC) measures, white blood cells count (WBC) with differential and leucocytes ratio (LR), serum levels of C-reactive protein (CRP), HDL cholesterol (HDL-C), triglycerides (TG), fasting glucose (fGlc), fecal microbiota α-diversity and the level of fecal SCFAs. The tertiary outcome measure will be the level of inflammation markers (interleukin-6 (Il-6) and tumor necrosis factor alpha (TNFα)) and oxidative stress parameters (total antioxidant capacity (TAC) and malondialdehyde (MDA)) in the blood serum (future use of biological material) (Table 2).

#### 2.4.1. Questionnaires and Scales

**Study Questionnaire (SQ)** has been constructed to collect basic information concerning sociodemographic and health-related data. The participants will provide information on personally identifying data, as well as known factors influencing microbiota composition and function: diet, smoking cigarettes, physical activity, overall health status and specific somatic diseases connected with dysbiosis, and the use of dietary supplements and medications known to influence microbiota to a large extent, e.g., antibiotics, proton-pump inhibitors (PPIs), metformin, laxatives, systemic steroids or nonsteroidal anti-inflammatory drugs (NSAIDs) or antipsychotics additionally known to induce both metabolic and inflammation alteration [44,45,46,47,48,49,50,51,52,53,54]. Both SQ and MQ applied every 2 weeks are based on exclusion criteria (Table 1). As the collected or processed data will belong to the special category, personal data will be pseudonymized.

As gut microbiota function is associated with self-reported long-term dietary patterns, the protocol includes a diet assessment of based on the **Food Frequency Questionnaire (FFQ) [55]**. We will use Polish self-administered FFQ-6 developed by Wądołowska [56]. FFQ will assess consumption of different food groups. The participants will be asked to estimate their “usual consumption” over the past year. The investigator administering the FFQ will invite the participants to describe their “usual” diet, rather than focus on their latest dietary consumption. Cross-check questions will be used to correct misreporting of fruit and vegetables as these tend to be overreported. Based on FFQ, the Healthy-Eating Index (HEI), which has been validated for studying the microbiome [55], will be calculated for each study participant.

The **Montgomery-Åsberg Depression Rating Scale (MADRS)** [57] was developed to assess depressive symptom severity in the investigation of treatment response. It consists of ten items chosen based on their ability to detect depression change, demonstrates psychometric properties equal to other standardized measures of depression, and is one of the most extensively used clinician-rated scales of depressive severity in clinical research [58]. We decided on the cut-off value of 13 points based on the clinical utility study by Duarte [59].

The **Depression, Anxiety, Stress Scales (****DASS)** is a set of three self-report scales designed to measure the negative emotional states of depression, anxiety, and stress [60]. Each of the three DASS scales contains 14 items, divided into subscales of 2–5 items with similar content. Subjects are asked to use 4-point severity/frequency scales to rate the extent to which they have experienced each state over the past week. 

**WHO Quality of Life-BREF (WHOQOL-BREF) instrument** [61] will be used to measure quality of life. It is a person-centered, multilingual, easily administered tool for subjective assessment, and it is designed for use in a wide range of diseases and conditions in both clinical settings and clinical trials.

**The Tool for Assessment of Suicide Risk (TASR)** [62] was designed to be used by clinicians to document their assessment of a patient who may be suicidal. TASR may be used to ensure that the most pertinent individual, symptom, and interview details necessary for the assessment of suicide risk have been addressed by the clinician and may be a “bedside” tool that helps the clinician determine the risk for suicide. We will use TASR to exclude severe suicide risk.

Although we will apply several questionnaires that will be self-administered, the study protocol includes an option for responses to be reviewed and any queries clarified in a face-to-face, e-mail, or telephone interview. To ensure that an FFQ is acceptable and understood by the subjects of the study it will be explained first. Clear instructions will be given at the beginning of the questionnaire and supported by relevant examples to resolve any doubts. Missing data on self-reported questionnaires will be treated in two ways. Questionnaires with more than 25% of incomplete questions will be excluded. In the case of questionnaires not exceeding the limit for incomplete data, an average value for the population will be substituted.

#### 2.4.2. Biological Samples

##### Venous Blood

Venous blood will be collected by qualified nurses according to the reliable protocols of collection, transport, and storage of biological material. Twenty milliliters (20 mL) of whole arm vein blood will be collected from each person in all the groups. Samples of fasting blood will be collected from the subjects after overnight rest, in the morning, between 8:00 and 10:00 a.m. 

Complete blood count (CBC), C-reactive protein (CRP), high-density protein cholesterol (HDL-c), TG, and fGlc will be measured subsequently. To obtain serum for future analysis (TAC, MDA, Il-6, TNF-α), blood will be transferred to the sterile tubes without an anticoagulant (on the so-called clot) and left at room temperature (approximately 30–45 min) to form a clot. After centrifugation at 1000× *g* (2400 rpm) for 10 min, the serum (supernatant) will be carefully separated from the clot into Eppendorf tubes. Approximately 2 mL of serum should be obtained from each patient at each time point. Until further analysis is made, the blood samples will be preserved in pyrogen/endotoxin-free collecting Eppendorf tubes (in 250–500 μL aliquots to avoid repeated freeze-thaw cycles) and stored frozen at −80 °C.


**Inflammation and oxidative stress parameters**


The serum obtained from the patients representing all the groups (PRO-DMS, PLC-DMS, PRO-D, and PLC-D) will be used to assess the inflammation protein level and oxidative stress parameters using enzyme-linked immunosorbent assays (ELISA) with the use of the highly specific antibodies. Small aliquots of serum (**100 or 200 μL per patient**) will be used in **each** ELISA assay. The intensity of the final colorimetric reaction, in proportion to the amount of protein/substrate bound, will be measured using the plate reader (ELx800, BioTek, Winooski, VT, USA) at 450 nm. The obtained results will be compared to the standard solution of known concentrations (specific for individual ELISA kits).

Based on ELISA, the following inflammation and OxS parameters will be analyzed:**Interleukin-6** (IL-6) is a multi-functional cytokine that regulates immune responses, acute phase reactions and hematopoiesis and may play a central role in host defense mechanisms [63]. It acts on a wide range of tissues, exerting growth-induction, growth-inhibition, and differentiation, respectively, depending on the nature of the target cells [46,47,48]. We plan to use the Human IL-6 ELISA Kit (Diaclone, Besançon, France, 950.030.192, minimum detectable dose of 2 pg/mL) or Interleukin-6 Human ELISA Kit (Biovendor, Brno, Czech Republic, RD194015200R). A highly specific capture antibody against IL-6 is coated to the wells of the microtiter strip plate provided in the kit.**Tumor necrosis factor alpha** (TNFα) is a polypeptide cytokine produced by monocytes and macrophages. It functions as a multipotent modulator of immune response and further acts as a potent pyrogen. TNFα circulates throughout the body responding to stimuli (infectious agents or tissue injury), activating neutrophils, altering the properties of vascular endothelial cells, regulating metabolic activities of other tissues, as well as exhibiting tumoricidal activity by inducing localized blood clotting [64,65,66].We plan to use the Human TNF alpha ELISA Kit (Diaclone, 950.090.192, minimum detectable dose of 8 pg/mL) or TNF-alpha Human ELISA, High Sensitivity Kit (Biovendor, RAF145R). A highly specific capture antibody against TNFα is coated to the wells of the microtiter strip plate provided in the kit.**Malondialdehyde** (MDA) is a naturally occurring product resulting from lipid peroxidation of polyunsaturated fatty acids. It is also produced in the prominent product in thromboxane A2 biosynthesis wherein cyclooxygenase 1 or cycloxygenase 2 metabolizes arachidonic acid to prostaglandin H2. MDA has a mutagenic and carcinogenic effect [67]. We plan to use the Highly Sensitive ELISA Kit for Malondialdehyde (Cloud Clone Corp, Houston, TX, USA. HEA597Ge, minimum detectable dose of 4.94 ng/mL). A highly specific capture antibody against MDA is coated to the wells of the microtiter strip plate provided in the kit.**Total antioxidant capacity** (TAC) (Total antioxidative status—TAS) is an analyte frequently used to assess the antioxidant status of biological samples and can evaluate the antioxidant response against the free radicals produced in the body in a given disease or analyzed condition. Overproductions of radical oxygen species (ROS) or insufficient defense mechanisms lead to a dangerous disbalance in the organism observed in lipid peroxidation, a mutagenic effect on DNA. The elevated level of ROS is associated with pathomechanisms implicated in aging and over 100 human diseases, e.g., cardiovascular disease, cancer, diabetes mellitus, inflammatory disease [68,69]. TAC measurements provide a tool for establishing links between antioxidant capacity and the risk of disease, as well as for monitoring of antioxidant therapy.

We plan to use the ImAnOx (TAS) Antioxidative Capacity Kit (Immundiagnostik, Bensheim, Germany, KC5200, minimum detectable dose of 130 μmol/L).

The estimated reference values of TAC (Based on Immunodiagnostik studies of ethylenediaminetetraacetic acid (EDTA)-plasma and serum of apparently healthy persons, *n* = 69):low antioxidative capacity < 280 μmol/Lmiddle antioxidative capacity 280–320 μmol/Lhigh antioxidative capacity > 320 μmol/L


**Detection of the IL-6 and TNFα in the serum**


A highly specific capture of antibodies against IL-6 or TNFα are coated to the wells of the microtiter strip plate provided in the kits. The sample (serum) is added to the wells in the plate. Binding of IL-6 or TNFα in the samples and known standards to the biotinylated antibodies is followed by subsequent binding of the secondary antibody (Streptavidin with horseradish peroxidase). Any excess unbound analyte and secondary antibody are removed. Next, the HRP conjugate solution is added, after incubation excess conjugate is removed. A chromogen substrate—TMB (3,3′,5,5′-Tetramethylbenzidine) is added to the wells resulting in a progressive development of a blue colored complex with the conjugate. The color development is then blocked by addition of acid turning the resultant final product yellow. The intensity of the produced colored complex is directly proportional to the concentration of IL-6 or TNFα present in the samples and standards.


**Detection of MDA in the serum**


The monoclonal antibody specific to malondialdehyde is pre-coated onto a microplate. A competitive inhibition reaction is launched between biotin labeled malondialdehyde and unlabeled malondialdehyde (standards or samples) with the pre-coated antibody specific to MAC. After incubation, the unbound conjugate is washed off. Next, avidin conjugated to horseradish peroxidase (HRP) is added to each microplate well and incubated. The amount of bound HRP conjugate is reverse proportional to the concentration of malondialdehyde in the sample.


**Determination of TAC in the serum**


The antioxidative capacity is determined by the reaction of antioxidants present in the sample with a defined amount of exogenously provided hydrogen peroxide (H_2_O_2_). During the incubation, H_2_O_2_ generates reaction products that absorb at 450 nm. Due to this effect and self-absorption, each sample has to be measured in two different conditions (with and without addition of enzyme). The antioxidants in the sample eliminate a certain amount of the provided hydrogen peroxide. The residual H_2_O_2_ is determined photometrically by an enzymatic reaction that involves the conversion of TMB to a colored product. After addition of a stop solution, the samples are measured at 450 nm in a microtiter plate reader. The difference in the values of the samples (those with and without the enzyme) is inversely proportional to the antioxidative capacity. 

Characteristics of the ELISA assays used to assess the inflammation and OxS parameters are described in Appendix B.

##### Feces

Samples of biological material, i.e., feces, will be taken by the patients themselves. Every patient entering the survey will be asked to collect a stool sample twice in the amount of 1 g. The study participants will be advised not to use laxatives, eat synthetic fat substitutes, or take fat-blocking nutritional supplements. It will be recommended to wait 48 h after potential barium enema introduction. The patients will be sampling feces after overnight fasting to establish a common reference point for food intake, a factor determining SCFAs synthesis. Following collection, the participants will be asked to deliver the sample as soon as possible to the study investigators. After delivery, the samples will be stored at −80 °C in the BioBank of the Medical University of Lodz until the analyses are performed. 


**SCFAs**


The following SCFAs will be evaluated: acetic acid (C 2:0), propionic acid (C 3:0), isobutyric acid (C 4:0 i), butyric acid (C 4:0 n), isovaleric acid (C 5:0 i), valeric acid (C 5:0 n), isocaproic acid (C 6:0 i), caproic acid (C 6:0 n), and heptanoic acid (C 7:0). SCFAs will be isolated from feces according to the Zhao method (modified by Roediger). Approximately 0.5 g of feces will be diluted in 5 mL of sterile water and homogenized for five minutes using the hand homogenizer. The pH will be adjusted to pH 2–3 using the 5 M HCl. The samples will be vortexed for ten minutes and centrifuged at 5000× *g* rpm for 20 min. The clear supernatant (after filtration using the Ø 400 µm filter) will be transferred to a chromatographic vial and will be analyzed with gas chromatography (GC) using the Agilent Technologies 1260 A GC system with a flame ionization detector (FID) and a fused-silica capillary column with a free fatty acid phase (DB-FFAP, 30 m, 0.53 mm, 0.5 um). SCFAs will be qualitatively identified by comparing the retention times to the standard, i.e., 2-ethylbuthyric acid. For the quantitative analysis, ChemStation Software (Agilent Technologies, UK) will be used. The measurement will be performed at the Department of Human Nutrition and Metabolomics, Pomeranian Medical University, Szczecin.


**Microbiota composition**


To characterize the microbial composition in the patients and identify changes over the course of the trial, we will perform the sequencing of the hypervariable region (V3 and V4 regions) of the 16S ribosomal RNA (rRNA) gene and fungal ITS1 (internal transcribed spacer) region. The procedure will be divided into four stages: 1. DNA isolation, quality and quantity assessment; 2. DNA 16S (V3-V4) library preparation with library validation; 3. next-generation sequencing (NGS) on the Illumina (San Diego, CA, USA) platform of 16S V3-4 rRNA, paired-end 2 × 250 bp with .fastq files generation; 4. bioinformatic analysis and report preparation. The bacterial and fungal DNA will be extracted from fecal samples with the use of the dedicated kit, followed by the DNA quality and quantity assessment. To assess the diversity of the microbiota in each patient, the bacterial and fungal DNA of the 16S ribosomal RNA (rRNA) gene will be amplified using the set of universal primers targeting the hyper-variable V3-V4 regions (for the bacterial composition analysis) and ITS1 region (for the fungal composition analysis). After the PCR reaction, the quality and quantity of the amplified product will be assessed, then the sequencing libraries will be prepared. After amplification of the libraries (for each patient and in each time point) the libraries will be sequenced on the Illumina platform (paired-end 2 × 250 bp with .fastq files generation). Finally, the reads—the 16S rRNA gene sequences/ITS1 region—will be clustered into operational taxonomic units (OTUs) at a similarity cut-off value of 97%. Microbiota gene sequencing will be performed by Bionanopark, Lodz.

Output reads will be analyzed using the Qiime2 platform [70]. For each sample dataset, the same quality check and filtration criteria will be applied. The read quality will be checked using FastQC software and statistics will be performed in Qiime2. The amplicon errors and chimeric sequences will be corrected by denoising using the q2:dada2 package. Following, the sequences will be clustered into operational taxonomic units (OTU) based on sequence similarity within the reads. The OTU sequences will be aligned based on GreenGenes database. An OTU table will be constructed representing the abundance of each OTU in each microbial sample. An α-diversity analysis will be performed by measuring the following indexes: Shannon, Simpson, Chao1, Abundance-based Coverage Estimator (ACE). It will be presented as a box-and-whisker plot for each index. A beta diversity analysis will be performed on the samples grouped by their origin. Weighted Unifrac and principal coordinate analysis (PCoA) will be measured to display microbiome distances between the samples. The results of the bacterial taxonomic analyses will be presented at the genus level as a heatmap. Additionally, multivariate ANOVA with permutations (PERMANOVA) will be used to test differences between the groups. 

### 2.5. Data Management

The study will use both self-administered and specialist-administered questionnaires filled in as face-to-face, paper-and-pencil, or on-line administration. The subjects will have an opportunity to review responses and clarify queries face-to-face, in an online meeting, or by e-mail or telephone interview. To make sure that questionnaires are acceptable and understood by the subjects of the study, they will be explained first with clear instructions and relevant examples.

Collected samples will include blood samples collected by qualified nurses, whereas feces samples will be collected by the study participants themselves. Data entry will be validated by the principal study investigator, and data review will be performed by the co-investigators.

The data will be organized as follows: 1. dataset: questionnaires filled in in the Microsoft Word (Microsoft Corporation, Redmond, Washington, USA) (.docx) format directly converted to PDF format; 2. dataset: description of full psychiatric examination. 3. dataset: laboratory findings (HDL-C, TG, fGlc, CBC, CRP, TAC, MDA, TNF-alpha, Il-6 in blood, MC digital results with bioinformatics analysis results in a graphic form, SCFAs concentration in feces) provided by laboratories in PDF format files. All the data will be summarized in the Microsoft Excel format. The files will be named according to pseudonymization of the study participants. We plan to reuse the data in future research.

The data will be catalogued in a standardized way in compliance with the requirements of findability, accessibility, interoperability, and reusability (FAIR) standards. The data will be physically stored in the office of principal study investigator (PSI) in paper and electronic form. PSI will use automatic backup services online ensuring protection against viruses by using anti-virus software, encrypted, and protected by passwords so that access to the data will be restricted. Selected data will be facilitated by open research data repository Zenodo without disclosure of the personal data. The metadata will be based on a generalized metadata schema used in Zenodo, which includes the title, creator (including ORCID number), date, contributor, subject, description, format, resource type, Digital Object Identifier (DOI), and access rights. A metadata description will be stored in JSON-LD format. 

According to the General Data Protection Regulation (EU) 2016/679 (GDPR), a study participant will provide informed consent to data processing for one or more purposes. To protect all the patients, the only way to identify an individual patient’s data will be by looking for his/her specific ID code printed in the informed consent.

CC-BY-NC-ND 4.0 type of creative common license schema will be applied.

### 2.6. Ethics

Each patient will receive information on the purpose and structure of the investigation. Deliberate written and informed consent to take part in the study will be required. The data collected and samples will be pseudonymized for the purposes of future use, and the informed consent form will include information on the future use of samples, material, and personal data. The study will be conducted in accordance with the Declaration of Helsinki.

The study protocol has been approved by the Bioethics Committee of the Medical University in Lodz, Poland, on 15 December 2020 (reference number RNN/228/20/KE). 

### 2.7. Analyses

As regards microbiota composition, we will assess α-diversity (within-sample diversity; richness using the observed OTUs), β-diversity (diversity of microbial community structure), and taxa abundance. 

To verify the hypothesis that persons with depression and accompanying MetS are more prone (than those without MetS) to microbiota derangement post probiotic ingestion, repeated measures ANOVA and modelling the relationship between multivariate data and predictor variables will be used. A similar approach will be used to investigate hypothesized change in MADRS score, SCFA, and metabolic parameters before and after probiotic intervention. 

Differences regarding variables i.e., clinical symptom severity (according to MADRS and DASS scores), α-diversity indexes, and β-diversity between the groups will be tested using *t*-test or ANOVA. Correlations between symptom severity and relative abundances of phyla, genera, and species as well as metabolic, inflammation, and OxS parameters will be assessed using a correlation test. The level of significance for statistical tests will be 5%. All statistical analyses will be performed using Statistica 13.1 (StatSoft, Tulsa, OK, USA).

As for the inflammation and oxidative stress analysis, an ANOVA Kruskal–Wallis test or Mann-Whitney test (depending on the number of groups), and Newman–Keuls’ multiple comparison test will be used to compare the levels of protein/substrate immunoexpression (assessed in ELISA assays), SCFAs levels, microbiota, and MADRS score between the groups of the patients. Spearman’s rank correlation coefficient, Mann-Whitney test, and ANOVA Kruskal–Wallis tests will be performed in order to evaluate the relationship between the protein immunoexpression levels, SCFAs, and the patients’ metabolic parameters.

## 3. Discussion

Obviously, our study protocol has both strengths and limitations. First of all, taking study population criteria into account, we decided to incorporate the whole category of depressive disorders according to the upcoming ICD-11 as the first step of eligibility screen in our study. The new diagnosis in this classification, underlying its impact on patients’ everyday functioning and quality of life, is anxious depression [42]. Additionally, the ICD-11 includes the category of mixed depressive and anxiety disorder (MDAD) in this section because of its importance in primary care settings and because of evidence confirming its overlap with mood symptomatology [71]. Therefore, our inclusion criteria may provide the basis for performing a study of the real-life population disabled because of depressive symptoms. On the other hand, the population will not be as homogenous as it would be with patients with MDD included only. Furthermore, MetS—as a set of anthropometric, clinical, and metabolic abnormalities (obesity, insulin resistance, dyslipidemia and hypertension)—has had different definitions developed, and some of the cut-offs of its criteria vary. In our research, we decided to use the International Diabetes Federation (IDF) definition and criteria because they address both clinical and research needs, providing a tool suitable for worldwide use [43].

Our microbiome may be modulated by numerous circumstances, especially diet [72]. Consumption of complex carbohydrates rich in dietary fibers—both the quantity and type of fat, as well as the type and amount of proteins in the diet—modulate the Firmicutes/Bacteroides ratio in the gut, having substantial effects on the gut microbiota [50]. Another strength of our study is to use self-reported measures that are validated for Polish conditions, such as the FFQ to assess dietary factors, and then to use those measures in statistical analysis. Apart from diet, there are many other factors influencing our microbiome, such as the mode of delivery (vaginal or caesarean), the mode of feeding in early infancy (breastfeeding vs. formula), exposure to viral or bacterial antigens, treatment with antibiotics and several other groups of pharmacotherapeutics, genetic predispositions, age, and environmental factors such as air pollution, exercise, or stress [44,48,49,51,73,74]. The factors that may influence the microbiota changes in the course of our study are included in our original questionnaires: the SQ at the beginning of the protocol and in the follow-up visit and the MQ in the monitoring process included at the V2 visit. Unfortunately, for obvious reasons we are not able to control most of environmental factors influencing the microbiota of the study subjects, such as air pollution, stressful life events, or genetics.

It is well-known that the probiotic effects are strongly strain-dependent. We will supplement the probiotic mixture of bacterial strains *Lactobacillus helveticus* Rosell^®^-52 and *Bifidobacterium longum* Rosell^®^-175. The above strains have been studied in experimental and animal models [75,76,77,78,79] as well as a few clinical studies assessing the functioning of microbiota–gut–brain axis, both in healthy subjects and depressed patients [80,81,82,83]. Moreover, in an animal model of MetS elevations in tumor necrosis factor alpha (TNF-α), interleukin (IL)-1β, IL-6, and IL-10, as well as expression of IL-6 mRNA, were attenuated in *L. helveticus*-treated rats [84]. Additionally, recent evidence suggests that intestinal Bifidobacterium species (spp.) positively correlates with improved insulin resistance and obesity, and this might be linked to metabolic inflammation [85].

It should be emphasized that different individuals can have taxonomically different but functionally similar microbiota [86], which makes functional assessment of microbiome health important. SCFAs are the most representative metabolites of fiber anaerobic fermentation [87] and also have the ability to diffuse across the blood–brain barrier [88]. There is evidence that an SCFA receptor—namely, free fatty acid receptor 3 FFAR3—is expressed in the brain tissue and sympathetic ganglia of rodents and was proved to act on the periportal afferent neural system and peripheral central nervous areas. Likewise, a human cerebrovascular endothelial cell line expressing FFAR3 lost its integrity post lipopolysaccharide (LPS) exposure, which was abolished after adding SCFAs (predominantly propionate) to the in vitro culture. Interestingly, a depletion of SCFAs was reported in MDD patients [89], and their administration was shown to alleviate symptoms of depression in mice [90]. Therefore, the analysis of the SCFAs concentration could serve as an indirect way to analyze microbiota composition, especially in the neuropsychiatric population. Given that bacterial functions are conserved across taxonomic groups, incorporating microbial function biomarkers such as SCFAs may be more productive than a purely taxonomic approach for understanding of the microbiome in depression. Based on the above, we decided to assess fecal microbiota function through measurement of SCFAs concentrations concurrently with microbiota composition. 

In conclusion, depression is one of the most disabling chronic diseases, and it impacts multiple domains of functioning. Additionally, depressive disorders often coexist with MetS, further increasing mortality risks. Current treatments for both depression and MetS remain suboptimal for many patients, thus making improvements and advances in the intervention options in great demand. Whilst probiotics may have benefits for some individuals who do not fully respond to antidepressant medications, the target clinical sample for this intervention is not fully recognized. This clinical trial will investigate the influence of probiotic supplementation on depressive and anxiety symptoms in patients with depression depending on metabolic syndrome comorbidity. In addition, this trial will assess intestinal microbiota function and composition, inflammatory status markers, and oxidative stress parameters as potential biomarkers of treatment efficacy. The trial, if successful, will establish a safe and easy-to-use treatment option (probiotic supplement) as an adjunctive therapy in patients only partially responsive to pharmacologic treatment. Given the personal and societal cost of treatment of chronic and widespread diseases such as depression and metabolic syndrome, the research may contribute to an advance that could improve the health care of many millions of people.

## Figures and Tables

**Figure 1 jcm-10-01342-f001:**
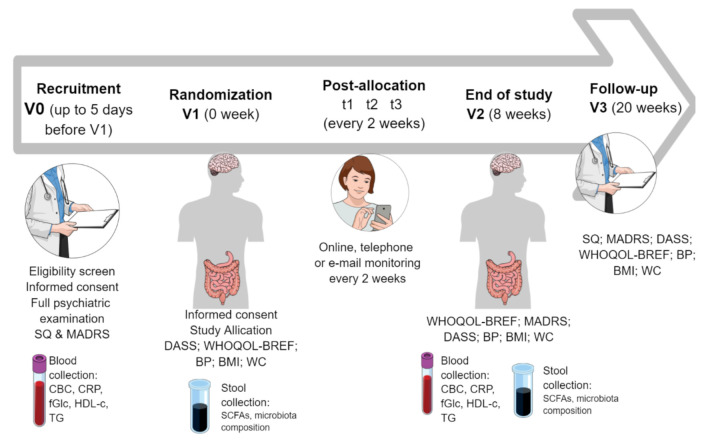
Overview of PRO-DEMET study timeline. Abbreviations: BMI: body mass index; BP: blood pressure; CBC: complete blood count; CRP: C-reactive protein; DASS: Depression, Anxiety, Stress Scale; fGlc: fasting glucose; HDL-c: high-density protein cholesterol; MADRS: Montgomery-Åsberg Depression Rating Scale; SCFAs: short-chain fatty acids; SQ: study questionnaire; TG: triglycerides; WC: waist circumference; WHOQOL-BREF: The World Health Organization quality of life-BREF.

**Figure 2 jcm-10-01342-f002:**
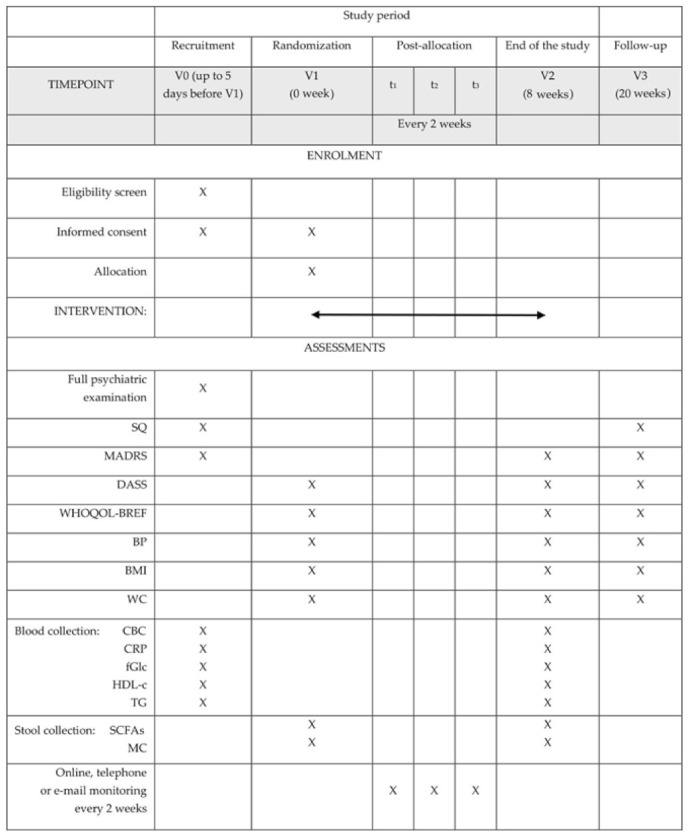
The schedule of enrolment, interventions, and assessments. Abbreviations: BMI: body mass index; BP: blood pressure; CBC: complete blood count; CRP: C-reactive protein; DASS: Depression, Anxiety, Stress Scale; fGlc: fasting glucose; HDL-c: high-density protein cholesterol; MADRS: Montgomery-Åsberg Depression Rating Scale; MC: microbiota composition; SCFAs: short-chain fatty acids; SQ: study questionnaire; TG: triglycerides; WC: waist circumference; WHOQOL-BREF: The World Health Organization quality of life-BREF.

**Table 1 jcm-10-01342-t001:** Eligibility criteria.

Inclusion criteria: Depressive disorders diagnosed according to International Classification of Diseases (ICD-11). Age above 18 years. MADRS score >= 13. Antidepressant and antianxiety medications or psychotherapy not changed three weeks prior to the beginning of the study. Depression + Metabolic syndrome (MetS) study groups: MetS diagnosed according to the International Diabetes Federation (IDF).
Exclusion criteria: Pregnancy. An infection and/or vaccination and/or treatment with antibiotics in the previous four weeks. Supplementation with probiotics or prebiotics in the previous four weeks. Being diagnosed with or having fresh symptoms of autoimmune, serious immunocompromised, inflammatory bowel diseases, cancer, IgE-dependent allergy in the previous four weeks. Current decompensated serious somatic disease. Psychiatric comorbidities (except for a specific personality disorder, an additional specific anxiety disorder, and caffeine or nicotine addiction). A major neurological disorder or any medical disability that may interfere with a subject’s ability to complete study procedures. A significant change in a dietary pattern in the previous four weeks. A significant change in dietary supplementation in the previous four weeks. A significant change in daily physical activity or an extreme sport activity in the previous four weeks. A significant change in a smoking pattern in the previous four weeks. A significant change in the treatment schema with proton-pump inhibitors (PPIs), metformin, laxatives, systemic steroids, nonsteroidal anti-inflammatory drugs (NSAIDs), antipsychotics, or any other medications influencing the microbiota according to present knowledge in the previous four weeks. High risk of suicide according to the Tool of Assessment of Suicide Risk (TASR). Current or recent participation in another research study involving an intervention that may alter outcomes that are relevant for this study. Any other condition or situation that, in the investigators’ opinion, would affect the compliance or safety of the individual involved. Depression study groups: MetS diagnosed according to the International Diabetes Federation (IDF).
Reasons for discontinuation of the study by a participant: Withdrawal of the informed consent. An infection and/or treatment with antibiotics during the course of the trial. Consuming any probiotics other than those studied during the course of the trial. Lack of compliance with the probiotic supplementation. Any change in the drug regimen during the study. Any exclusion criteria identified after the enrolment. Any serious adverse event occurring during the course of the trial.

**Table 2 jcm-10-01342-t002:** Outcome measures of PRO-DEMET study. Abbreviations: BMI: body mass index; BP: blood pressure; CRP: C-reactive protein; DASS: Depression, Anxiety, Stress Scale; FFQ: food frequency questionnaire; fGlc: fasting glucose; HDL-c: high-density protein cholesterol; Il-6: interleukin-6; LR: leucocytes ratio; MADRS: Montgomery-Åsberg Depression Rating Scale; MC: microbiota composition; MDA: malondialdehyde; MQ: monitoring questionnaire; SCFAs: short-chain fatty acids; SQ: study questionnaire; TAC: total antioxidant capacity; TASR: Tool for Assessment of Suicide Risk; TG: triglycerides; TNFα: tumor necrosis factor alpha; WBC: white blood cells; WC: waist circumference; WHOQOL-BREF: The World Health Organization quality of life-BREF.

Psychometric Tools		Physical Examination	Biological Samples			
Self-Administered	Administered by an Interviewer		Blood			Feces
			Metabolic Parameters	Inflammation Parameters	OxS Parameters	
SQ	MADRS	BP	HDL-C	WBC	TAC	MC
DASS	TASR	BMI	TG	LR	MDA	SCFAs
WHOQOL-BREF		WC	fGlc	CRP		
FFQ				Il-6		
MQ				TNFα		

## Data Availability

Data sharing not applicable.

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
