# Peer review of "The Influence of Probiotic Supplementation on Depressive Symptoms, Inflammation, and Oxidative Stress Parameters and Fecal Microbiota in Patients with Depression Depending on Metabolic Syndrome Comorbidity—PRO-DEMET Randomized Study Protocol"

_jcm, 2021, doi:10.3390/jcm10071342_

Round 1
Reviewer 1 Report
Here are few suggestions :
- There are typos, double spacing and with symbols like '??' which needs to be addressed.
- In the introduction, I would suggest give more detailed overview how critical are gut microbiota for depression and metabolic syndrome. You can give example change in gut microbiota profile. Also how in mouse model with alternation in gut microbiota, animals display depression like phenoype or metabolic condition. Treatment with probiotic the conditions were ameliorated.
- The study aims to look into patients with depression who later develop metabolic disorder. How will you take that into account while considering patients if they already had a metabolic syndrome before depression ? Or was the metabolic syndrome developed during the course of medication?
- Im sorry if I missed, but it was not mentioned will you keep a record if patient enrolled for the study had a history of metabolic disorders making them it more likely to develop metabolic syndrome.
-
Will you also keep a record on the kind of anti-psychotics patients were one and if they will continue to be on anti-psychotics ? Its critical because some anti-psychotics have been shown to interfere with gut microbiota so there is a possibility they might impair or interfer with probiotic as well.
- Apart from excluding patients on laxatives, will you also be keeping a record of gastrointestinal conditions in the enrolled patients. As gastrointestinal condition is often reported in patient with depresssion and metabolic syndrome. It would be useful to keep an account at start and over the course to see if the conditions also improve with treatment with probiotics.
- You mentioned in the sections faeces will be collected but its not clearly mentioned how much of the biological material will be collected from the patients. As this protocol will be used in near future so it would be useful for others if you may mention this information.
Author Response
Dear Reviewer,
My point-by-point response below.
- There are typos, double spacing and with symbols like '??' which needs to be addressed.
I have tried to correct all the typos.
- In the introduction, I would suggest give more detailed overview how critical are gut microbiota for depression and metabolic syndrome. You can give example change in gut microbiota profile. Also how in mouse model with alternation in gut microbiota, animals display depression like phenoype or metabolic condition. Treatment with probiotic the conditions were ameliorated.
I have improved the introduction, gave more detail on the changes of microbiota in depression and obesity, mainly based on recently published meta-analyses. Animal models studies have also been included, as well as more details on probiotics efficacy in civilization diseases. I used the "Track Changes" function in Microsoft Word.
- The study aims to look into patients with depression who later develop metabolic disorder. How will you take that into account while considering patients if they already had a metabolic syndrome before depression ? Or was the metabolic syndrome developed during the course of medication?
and
- Im sorry if I missed, but it was not mentioned will you keep a record if patient enrolled for the study had a history of metabolic disorders making them it more likely to develop metabolic syndrome.
Pro-demet study aims to investigate 2 distinct groups of patients: depressive ones without metabolic syndrome and depressive patients with comorbid metabolic syndrome. We will assess probiotics efficacy depending on metabolic syndrome comorbidity. The presence of the syndrome will be established on V1 visit (as well as V2 visit) according to IDF criteria and we will stratify randomization according to metabolic syndrome presence.
All the medications that may cause metabolic abnormalities will be considered.
- Will you also keep a record on the kind of anti-psychotics patients were one and if they will continue to be on anti-psychotics ? Its critical because some anti-psychotics have been shown to interfere with gut microbiota so there is a possibility they might impair or interfer with probiotic as well. We will keep record of all the medications taken by the study subjects, including antipsychotics. One of our inclusion criterium is “A significant change in the treatment schema with proton-pump inhibitors (PPIs), metformin, laxatives, systemic steroids, nonsteroidal anti-inflammatory drugs (NSAIDs), antipsychotics or any other medications influencing the microbiota according to present knowledge in the previous four weeks.” It also applies to intervention period. More details on factors influencing microbiota and included in our study eligibility screen may be found in chapter “4.1. Questionnaires and scales”.
- Apart from excluding patients on laxatives, will you also be keeping a record of gastrointestinal conditions in the enrolled patients. As gastrointestinal condition is often reported in patient with depresssion and metabolic syndrome. It would be useful to keep an account at start and over the course to see if the conditions also improve with treatment with probiotics.
Thank you for your remark. We plan to follow somatic symptoms according to subjects reports. We also have exclusion criterium “Being diagnosed with or having fresh symptoms of autoimmune, serious immunocompromised, inflammatory bowel diseases, cancer, IgE-dependent allergy in the previous four weeks. “ which applies to intervention period as well.
- You mentioned in the sections faeces will be collected but its not clearly mentioned how much of the biological material will be collected from the patients. As this protocol will be used in near future so it would be useful for others if you may mention this information.
I add the information. It will be 1 gram of faeces at the beginning and the end of the study.
I also want to mention that we have changed eligibility criteria a little bit considering reviewers’ remarks as well as the most recent publications on microbiota.
Additionally, to ease understanding of the material we have added another table (Tabl. 2 Outcome measures of PRO-DEMET study), a study timeline graphic (Fig. 1 Overview of Pro-demet study timeline) and a graphical abstract.
Sincerely,
Authors.

Reviewer 2 Report
As it stands, presented manuscript describes the protocol of future study regarding the influence of probiotic supplementation on patients with Depression Depending on Metabolic Syndrome Comorbidit.
It is hard to estimate scientific soundness of the article since it is only proposed study. First of all abstract should be rewritten to clearly state that fact. As it is - impression is made that this study has already been performed.
If editor would decide to accept this manuscript for publication, following edits are highly recommended:
- Graphical representation of the material will ease understanding of the material. Diagram/charts will be highly appreciated.
- Consider proof-reading for English language and checking text for formatting. Consider highlighting in bold paragraph subtitles (e.g. in section 2.4.2 highlight "SCFS", "Microbiota composition", etc. Same in other sections)
- Lack of modern literature:
Consider including following publications on the topic of probiotic supplementation in your references list:
https://doi.org/10.3390/microorganisms8081225 (2020)
https://doi.org/10.14814/phy2.14610 (2020)
https://doi.org/10.1186/s12866-021-02099-0 (2021)
https://doi.org/10.3390/nu12030605 (2020)
Best wishes in your research
Author Response
Please see the attachment.

This manuscript is a resubmission of an earlier submission. The following is a list of the peer review reports and author responses from that submission.